# An automated framework for exploring and learning potential-energy surfaces

Yuanbin Liu [1], Joe D. Morrow[1], Christina Ertural [2], Natascia L. Fragapane [1], John L. A. Gardner[1], Aakash A. Naik [2,3], Yuxing Zhou [1], Janine George [2,3] ✉ & Volker L. Deringer [1] ✉

Machine learning has become ubiquitous in materials modelling and now routinely enables large-scale atomistic simulations with quantum-mechanical accuracy. However, developing machine-learned interatomic potentials requires high-quality training data, and the manual generation and curation of such data can be a major bottleneck. Here, we introduce an automated framework for the exploration and fitting of potential-energy surfaces, implemented in an openly available software package that we call `autoplex` ('automatic potential-landscape explorer'). We discuss design choices, particularly the interoperability with existing software architectures, and the ability for the end user to easily use the computational workflows provided. We show wide-ranging capability demonstrations: for the titanium–oxygen system, $SiO_2$, crystalline and liquid water, as well as phase-change memory materials. More generally, our study illustrates how automation can speed up atomistic machine learning in computational materials science.

Machine-learned interatomic potentials (MLIPs) are now established as the method of choice for large-scale, quantum-mechanically accurate atomistic simulations[1–5], with applications ranging from high-pressure research[6–8] to the discovery of molecular reaction mechanisms[9,10] and even to the realistic modelling of proteins[11]. MLIPs are trained on quantum-mechanical reference data—typically derived from density-functional theory (DFT)—using a variety of methods from linear fits[12–14] and Gaussian process regression[15] to neural-network architectures[16–22]. Traditionally, MLIPs have been largely hand-crafted models, built using configurations manually tailored for domain-specific tasks[23–26], such as the fracture of silicon[24] or the crystallisation of Ge–Sb–Te memory materials[26]. More recently, a trend has emerged towards pre-trained or 'foundational' MLIPs[27,28]: these models are fitted to large datasets including many chemical elements, and can be fine-tuned for downstream tasks[27,29].

With sophisticated MLIP fitting frameworks available and continuously improving, we argue that the next area of innovation lies in the data used to train the models[30,31]. The aforementioned fine-tuning is one example of the more general challenges in this field:

constructing high-quality datasets still remains a non-trivial, time- and labour-intensive aspect of MLIP model development, and (more) efficient methods for data generation are needed. Commonly, active-learning strategies are now used to iteratively optimise datasets by identifying rare events and selecting the most relevant configurations via suitable error estimates[32–34]. Active learning has been widely used to explore phase transitions[35–37] and chemical reactions[38–40]. And yet, such methods often still rely on costly ab initio molecular dynamics (MD) computations to expand and refine the training datasets.

Currently available 'foundational' MLIPs are typically fitted to a dataset comprising relaxation trajectories of diverse crystalline materials sourced from the Materials Project initiative[41,42]. The present work is concerned with the—somewhat orthogonal—question of how one can build an MLIP model from scratch: exploring local minima but also highly unfavourable regions of a given potential-energy surface, which need to be taught to a robust potential. Previous work showed that random structure searching (RSS) provides a particularly promising approach for the exploration and iterative fitting of configurational space (Fig. 1a–c)[43–49]. The principle of the original RSS approach,

[1]Inorganic Chemistry Laboratory, Department of Chemistry, University of Oxford, Oxford, UK. [2]Materials Chemistry Department, Federal Institute for Materials Research and Testing (BAM), Berlin, Germany. [3]Institute of Condensed Matter Theory and Solid-State Optics, Friedrich Schiller University Jena, Jena, Germany. ✉e-mail: janine.george@bam.de; volker.deringer@chem.ox.ac.uk

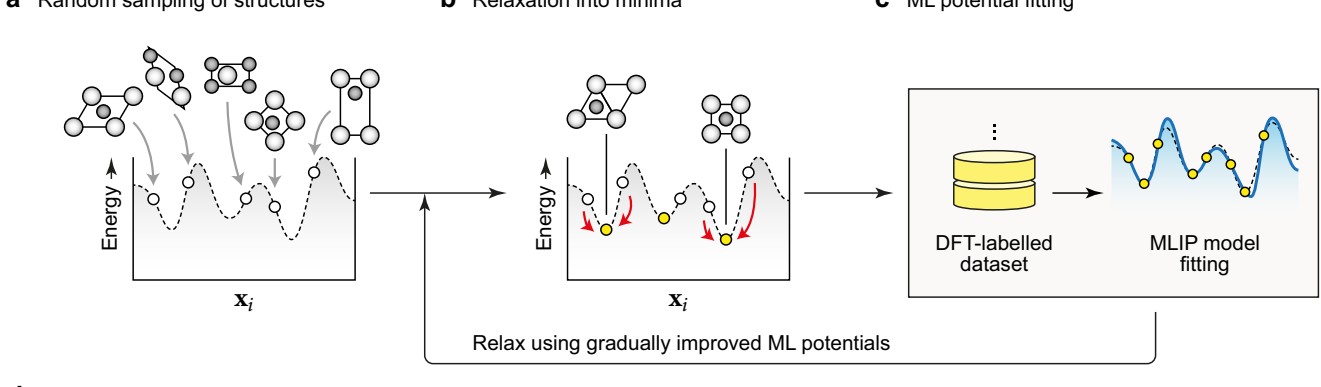

**a** Random sampling of structures    **b** Relaxation into minima    **c** ML potential fitting

Relax using gradually improved ML potentials

**d** Automation through workflows

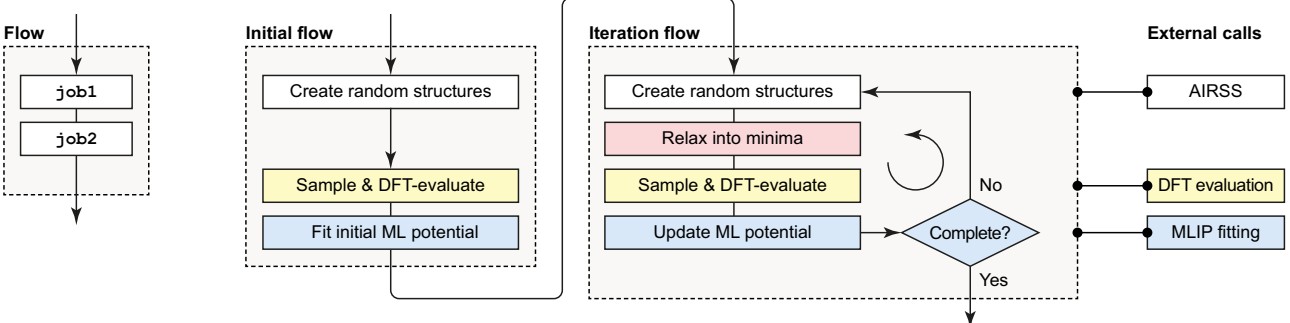

**Fig. 1 | Exploration and fitting of potential-energy surfaces.** From left to right, the process involves: (**a**) the random sampling of structures, using suitable constraints (in this simple cartoon, we assume that we are searching for crystal structures of a binary compound, containing one atom of each type under periodic boundary conditions, with $\mathbf{x}_i$ denoting the atomic coordinate vector of the $i$-th structure in configuration space); (**b**) the relaxation of those structures into local minima which correspond to relevant crystal structures; and (**c**) the fitting of machine-learned (ML) potentials. Panel (**d**) provides a high-level overview of the workflow structure: the left-hand side is a cartoon of a 'flow' consisting of two arbitrary 'jobs' to be carried out in sequence, and the schematic in the main part of the figure summarises our approach taken in `autoplex`. We colour-code the different key steps in white (random structure generation, here using the `buildcell` code of Ab Initio Random Structure Searching, AIRSS[50,51]), red (iterative exploration), yellow (sampling and single-point computations, the latter using density functional theory, DFT), and blue (fitting of a machine-learned interatomic potential, MLIP).

known as Ab Initio Random Structure Searching (AIRSS)[50,51], is illustrated by panels a–b in Fig. 1, and the approach proposed in ref. 43 unifies it with MLIP fitting: using gradually improved potential models to drive the searches, without relying on any first-principles relaxations (only requiring DFT single-point evaluations) or pre-existing force fields. We note that AIRSS has been used as part of the approach for developing the recently described 'graph networks for materials exploration' (GNoME)[52] and MatterSim[53] models to create structurally diverse training data.

To date, these RSS-related approaches still depend heavily on the user's expertise and time and are by no means trivial to implement. This challenge is particularly apparent for very large training datasets, where manually executing and monitoring tens of thousands of individual tasks is practically impossible. A similar challenge was previously observed in DFT-driven materials discovery, and automation approaches have been developed in response[54]: numerous workflow systems can now be used to streamline first-principles materials exploration[55–59]. Owing to such efforts, together with high-performance computing facilities, DFT-driven high-throughput simulations have become commonplace today and have played an important role in the computational discovery of new materials[60–62]. However, the same level of maturity has not yet been achieved for the full development pipeline of MLIPs (exploration, sampling, fitting, refinement): although important steps have recently been made[63–69], the development of fully automated workflows for MLIPs remains in high demand.

Here, we describe an automated implementation of iterative exploration and MLIP fitting through data-driven random structure searching, focusing on both the automation infrastructure and its implications for materials modelling applications. We show that, within the open-source `autoplex` code we have developed, MLIP fitting can be carried out in a largely automated way on high-performance computing systems and in a high-throughput manner—and we show how the resulting potentials are robust and useful, especially given the ease with which they can be created from scratch. Note that our benchmarks here are restricted to bulk systems, whereas surfaces, interfaces, and reaction pathways remain important and challenging frontiers for future extension. We expect that this work will contribute to the mainstream uptake of ML-driven atomistic modelling in the wider community in the years ahead.

## Results

### The `autoplex` framework

The `autoplex` framework is a modular set of software that is interfaced to widely-used computational and automation infrastructure where applicable. In particular, our code follows the core principles of (and reuses some functionalities implemented in) the `atomate2`[70] framework, which in turn underpins the Materials Project initiative[41,42]. The code is available openly via GitHub, distributed under a permissive licence, and accompanied by documentation to facilitate its uptake. In the present work, we mainly use the Gaussian approximation potential (GAP)[15] framework to drive exploration and potential fitting,

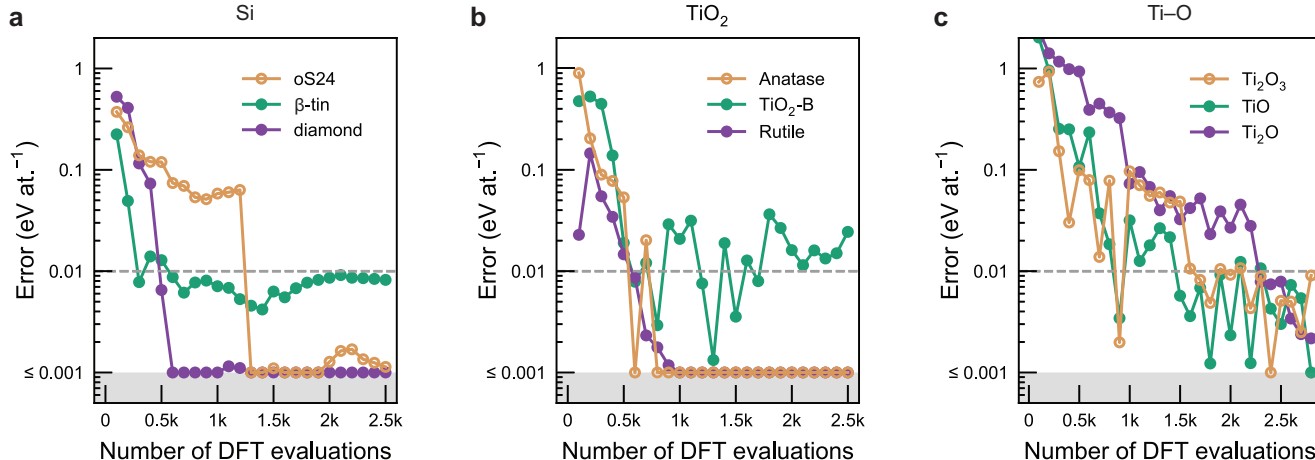

**Fig. 2 | Automated exploration of key materials systems.** We show the results of `autoplex` runs, characterised by the energy prediction error of the models, for gradually more complex scenarios: (**a**) the diamond-type and $\beta$-tin-type form of elemental silicon, as well as the open-framework oS24 allotrope described in ref. 71; (**b**) the common rutile and anatase forms of $TiO_2$, as well as the less common 'B' polymorph; and (**c**) three phases from the full binary Ti–O system. Note that the exploration for panel (**b**) included only stoichiometrically precise $TiO_2$ structures, whereas that in panel (**c**) used a variable composition range. The data are shown in a style similar to ref. 47. The grey-shaded area indicates errors below 0.001 eV at.$^{-1}$, which we take to roughly correspond to the accuracy limit achievable using density functional theory (DFT)[47]. The dashed line marks an error of 0.01 eV at.$^{-1}$. Source data are provided as a Source Data file.

leveraging the data efficiency of GAP and building on previous successful applications for this purpose[43,47]. We do note that `autoplex` is designed to accommodate other MLIP architectures as well.

We demonstrate the validity of the method using key examples, moving up in difficulty from elemental silicon to the polymorphs of $TiO_2$, and onwards to the full binary titanium–oxygen system. Figure 2 shows the evolution of energy prediction errors for relevant crystalline modifications with an increasing number of DFT single-point evaluations used to create GAP-RSS models. Each panel in Fig. 2 corresponds to a separate round of iterative, automated training using `autoplex` (details are provided in Supplementary Note 1). Each step encompasses 100 single-point DFT evaluations whose results are added to the training dataset. The root mean square error (RMSE) between the predicted and reference energies is used as a performance metric to evaluate models.

Silicon (Fig. 2a)—perhaps the classic test case for any materials simulation method—has a main allotrope with the diamond-type structure, and multiple higher-pressure forms, prominently the $\beta$-tin type structure. We also include the open-framework oS24 allotrope of silicon[71] as an example of a metastable phase that has been experimentally characterised and includes lower-symmetric atomic environments, as a more challenging test case for `autoplex`. The tests in Fig. 2a are similar to earlier work in ref. 47 and, for consistency, use the same DFT parameters as in that previous work (e.g., the same exchange–correlation functional). All three allotropes are well described to within an accuracy on the order of 0.01 eV at.$^{-1}$: the highly symmetric diamond- and $\beta$-tin-type structures with $\approx$ 500 DFT single-point evaluations, the oS24 structure within a few thousand (Fig. 2a). We take 0.01 eV at.$^{-1}$ to be a 'sensible' accuracy target for random exploration, and indicate this value by dashed horizontal lines in Fig. 2. The higher numerical error for $\beta$-tin-type compared to diamond-type silicon (green vs. purple in Fig. 2a) is consistent with previous work on a general-purpose GAP model for the element[24] as well as an earlier GAP-RSS study[47].

The binary oxide, $TiO_2$, is structurally highly diverse, therefore forming a suitable next target for testing a crystal-structure searching method. The compound mainly exists in two common forms—rutile and anatase—which contain distorted octahedrally coordinated $[TiO_6]$ units and differ in the connectivity of the coordination polyhedra. We also include the bronze-type ('B-') polymorph of $TiO_2$, which is less abundant, but has been of interest for battery research[72,73]. Figure 2b shows that while the two main polymorphs are again correctly recovered, and the prediction error for $TiO_2$-B reduces to a few tens of meV at.$^{-1}$ as well, the latter polymorph appears to be distinctly more difficult to 'learn' than the two simpler ones.

We finally study the exploration of a full binary system containing multiple phases with varied stoichiometric compositions. In Fig. 2c, we present the results of testing our approach on compounds with different stoichiometric compositions (and electronic structure), viz. $Ti_2O_3$, TiO, and $Ti_2O$. While we truncate the plot at 0.001 eV at.$^{-1}$, we emphasise that we already consider achieving an accuracy of $\approx 0.01$ eV at.$^{-1}$ to be sufficient in this test of random exploration. It is fair to observe that compared to simpler phases such as rutile and anatase, achieving the target accuracy in this case requires a greater number of iterations, as the search space is more complex. We explore the behaviour for RSS runs up to 5000 structures in Supplementary Fig. 1.

Table 1 shows results for relevant main Ti–O polymorphs, evaluated with the $TiO_2$ potential characterised in Fig. 2b, and with the Ti–O potential from Fig. 2c. This is a particularly instructive case because it allows us to probe the limits of the method: if only trained on $TiO_2$, a GAP-RSS model will faithfully capture the polymorphs with this specific stoichiometric composition, but produces unacceptable errors when applied to compositions that deviate largely from the stoichiometry ( >100 meV at.$^{-1}$ for one of the $Ti_3O_5$ polymorphs, and >1 eV at.$^{-1}$ for rocksalt-type TiO, for example). In contrast, by training the model for the full Ti–O system (cf. Fig. 2c), we are able to obtain an accurate description for several different phases. This example highlights the flexibility of `autoplex` in handling varying stoichiometric compositions, requiring no substantially greater effort from the user than that for a single stoichiometrically precise compound—all that is required is a change in input parameters for RSS, and probably a moderately increased amount of computational resources. We note that likely, the overall accuracy of those models can be improved further—for the time being, we report them as obtained with standard DFT and GAP fitting settings. An examination of potential data leakage is provided in Supplementary Note 1.

## High-level potentials at moderate computational cost

A distinct advantage of the RSS approach for potential fitting is that it requires only single-point computations to generate the reference data[43]. We therefore posit that we are able to use `autoplex` to easily build high-quality potentials beyond the 'standard' GGA functionals

that are commonly used. We show here an example where higher-level data are crucial.

The seemingly simple silicon dioxide, $SiO_2$, has long posed challenges for atomistic modelling (see ref. 25 and references therein). We start by running our workflow using the economic

### Table 1 | Comparison of root mean square errors (RMSEs) across major Ti–O polymorphs between the $TiO_2$-specific model (Fig. 2b) and the full Ti-O model (Fig. 2c), both based on Gaussian approximation potential (GAP) models constructed using random structure searching (RSS)

| Compound | Structure type | RMSE (meV at.$^{-1}$) | |
|---|---|---|---|
| | | GAP-RSS ($TiO_2$ only) | GAP-RSS (Full Ti–O system) |
| $TiO_2$ | Anatase | 0.1 | 0.7 |
| $TiO_2$ | Baddeleyite | 1.1 | 28 |
| $TiO_2$ | Brookite | 10 | 8.2 |
| $TiO_2$ | Columbite | 1.0 | 0.9 |
| $TiO_2$ | Rutile | 0.2 | 1.8 |
| $TiO_2$ | $TiO_2$-B | 24 | 20 |
| $Ti_3O_5$ | $Ti_3O_5$ | 105 | 19 |
| $Ti_3O_5$ | $V_3O_5$(HT) | 10 | 4.1 |
| $Ti_2O_3$ | $Al_2O_3$ | 144 | 9.1 |
| TiO | NaCl | —[a] | 0.6 |
| $Ti_2O$ | $Ti_2O$ | —[a] | 2.2 |
| $Ti_3O$ | $Ti_3O$ | —[a] | 23 |

Each error estimate is based on 10 rattled structures, generated by applying random displacements to the atomic positions of the ground-state structures with a standard deviation of 0.01 Å.
[a] In these cases, errors were > 1 eV at.$^{-1}$, and numerical values are therefore not meaningful to report.

Perdew–Burke–Ernzerhof (PBE) functional[74]. We then re-run the workflow with the same RSS parameter settings (Supplementary Note 1), this time using the Strongly Constrained and Appropriately Normed (SCAN) functional[75]. Figure 3a shows that for $\alpha$-quartz, achieving high prediction accuracy (1 meV at.$^{-1}$) with both PBE and SCAN functionals requires less than 10,000 CPU core hours, corresponding to nominal costs on the order of $100. For the structurally more complex $\alpha$-cristobalite polymorph, SCAN incurs higher computational costs compared to PBE but still remains at the scale of 10,000s of core hours.

Why SCAN? The importance of using this higher-rung functional in this case becomes apparent when inspecting the absolute energy predictions of the different MLIPs. PBE fails to correctly reproduce the stability ordering of $SiO_2$ polymorphs, incorrectly predicting that $\alpha$-cristobalite is more stable than $\alpha$-quartz (identified by a negative value in Fig. 3b, highlighted in orange). This is an issue that is well known for some DFT methods (see, e.g., an early study in ref. 76). By contrast, SCAN does identify $\alpha$-quartz as stable—and indeed this DFT level has been used to train an MLIP for $SiO_2$ before[25], as well as a more comprehensive one for the binary Si-O system[77]. Further work could now include a direct benchmarking of our candidate RSS-derived MLIPs, as well as other data-generation workflows[65–68], against the models of refs. 25 and 77.

Figure 3c illustrates the energy difference between $\alpha$-cristobalite and $\alpha$-quartz as a function of the number of training structures for GAP models trained with PBE and SCAN. The results indicate that the stability predictions from both GAP-RSS runs align well with the respective DFT results—but because of the underlying training data, the GAP@SCAN model is qualitatively correct ($\Delta E > 0$), whereas the GAP@PBE model is not ($\Delta E < 0$).

Table 2 shows the results for additional $SiO_2$ polymorphs: aside from $\alpha$-cristobalite, the stability order of moganite and tridymite with respect to $\alpha$-quartz is also inaccurately predicted by DFT@PBE and GAP@PBE. In contrast, DFT@SCAN and GAP@SCAN successfully capture the stability of different crystal structures with overall accuracy. However, in terms of numerical precision, there is no significant

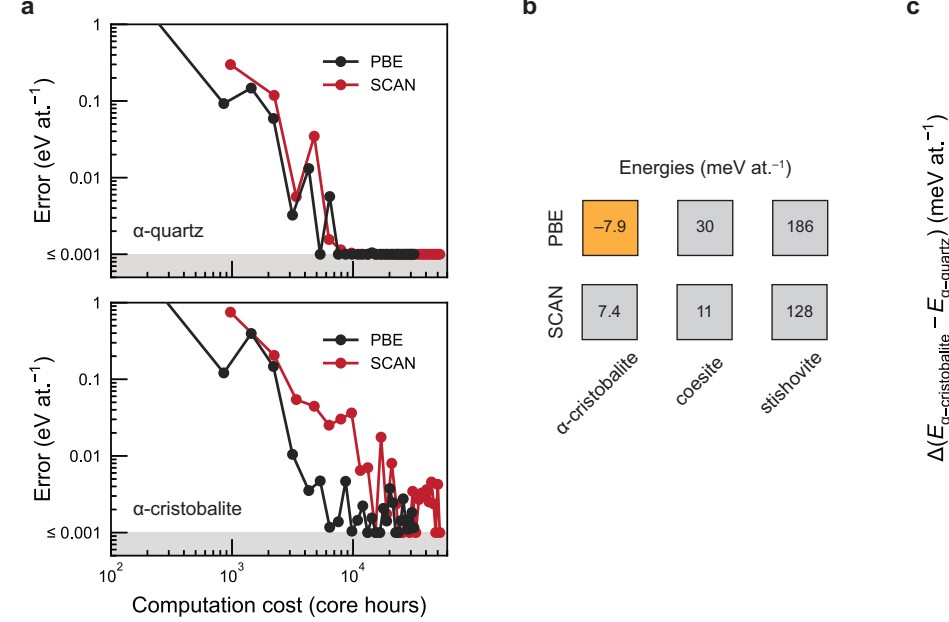

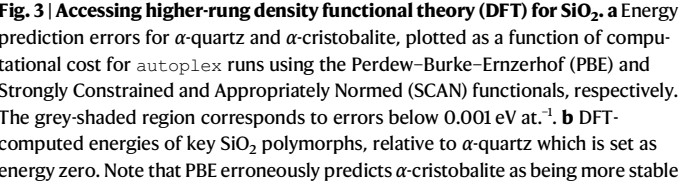

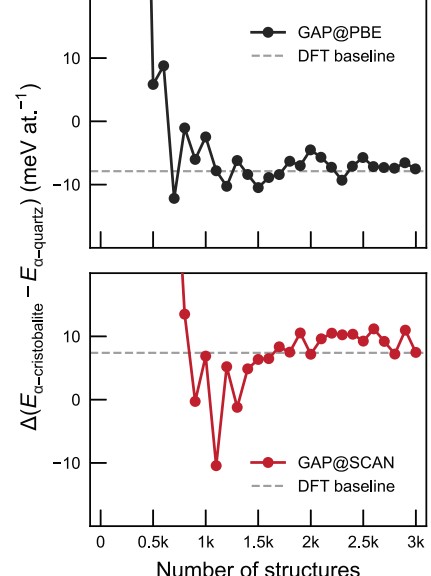

**Fig. 3 | Accessing higher-rung density functional theory (DFT) for $SiO_2$. a** Energy prediction errors for $\alpha$-quartz and $\alpha$-cristobalite, plotted as a function of computational cost for `autoplex` runs using the Perdew–Burke–Ernzerhof (PBE) and Strongly Constrained and Appropriately Normed (SCAN) functionals, respectively. The grey-shaded region corresponds to errors below 0.001 eV at.$^{-1}$. **b** DFT-computed energies of key $SiO_2$ polymorphs, relative to $\alpha$-quartz which is set as energy zero. Note that PBE erroneously predicts $\alpha$-cristobalite as being more stable

than $\alpha$-quartz; this incorrect prediction is highlighted by the orange shading. **c** Energy difference between $\alpha$-quartz and $\alpha$-cristobalite as a function of the number of training structures for Gaussian approximation potential (GAP) models trained with PBE (GAP@PBE) and SCAN (GAP@SCAN). The symbol $E$ denotes the energy. The dashed line represents the benchmark energy difference between $\alpha$-quartz and $\alpha$-cristobalite obtained from DFT calculations. Source data are provided as a Source Data file.

**Table 2 | Comparison of energy differences between various SiO$_2$ polymorphs and α-quartz using different density functional theory (DFT) methods and Gaussian approximation potential (GAP) models at the PBE and SCAN levels**

| Structure type | $\Delta(E - E_{\alpha\text{-quartz}})_{PBE}$ | | $\Delta(E - E_{\alpha\text{-quartz}})_{SCAN}$ | |
|---|---|---|---|---|
| | DFT | GAP-RSS | DFT | GAP-RSS |
| coesite | 30 | 31 | 11 | 12 |
| stishovite | 186 | 185 | 128 | 127 |
| α-cristobalite | −7.9[a] | −7.5[a] | 7.4 | 7.5 |
| moganite | −0.4[a] | −3.5[a] | 1.8 | 6.2 |
| tridymite | −8.2[a] | −11[a] | 8.4 | 7.6 |

The GAP models are generated using random structure searching (RSS). The symbol $E$ represents the energy. (Unit: meV at.$^{-1}$).

[a]These polymorphs are erroneously predicted to be more stable than α-quartz at the PBE level.

difference between the performance of the GAP@PBE and GAP@SCAN models. Even polymorphs with more complex unit cells are still predicted within our primary accuracy target of 10 meV at.$^{-1}$. An energy–volume analysis demonstrates that the structures generated through RSS iterations tend to approach the ground-state phases of SiO$_2$ polymorphs (Supplementary Fig. 2).

## Describing water with different architectures

Apart from crystalline inorganic solids, we assess the ability of our approach to describe condensed-phase molecular systems. Among them, water is particularly challenging, where, in addition to coexisting covalent bonds and hydrogen bonds, van der Waals forces also play non-negligible roles. Such interatomic complexity makes the modelling of water highly sensitive to the choice of DFT functionals (a systematic investigation can be found in ref. 78). Here, we train a GAP-RSS potential at the level of revPBE-D3 with zero-damping[79], which has been shown to reproduce the experimental radial distribution functions (RDFs) of liquid water with accuracy comparable to hybrid functionals[80,81], but at much lower computational cost. Beyond GAP, we explore the use of a different MLIP fitting architecture, focusing on graph-neural-network potentials, aiming to better capture the aforementioned interactions in different condensed phases of water.

Figure 4 characterises results for liquid water and ice polymorphs. We run GAP-RSS iterations as before, during which the model progressively identifies an increasing variety of hydrogen-bonded environments (Supplementary Fig. 3). The resulting potentials provide an overall correct description of the structure of liquid water in MD simulations (Fig. 4a), although the intensity of the first peak in the O−O RDF is more pronounced in the simulation than in experiment. We do not present the H···H interactions here, as our MD simulations do not account for nuclear quantum effects, which significantly influence the first peak of the RDF—see, e.g., ref. 82 for a discussion of these effects.

With an initial GAP-RSS-based dataset available, we carried out fits using the NequIP architecture[19], and used the resulting models to drive MD simulations in LAMMPS[83]. We used the implementations of the NequIP architecture, the training loop, and the relevant `pair style` from the `graph-pes` package[84] to do this (hyperparameters are provided in Supplementary Note 1). We find that both GAP and NequIP models fitted to the same GAP-RSS dataset can accurately describe key structural features of liquid water (Fig. 4a–b), with the NequIP model showing some improvements for the second peak in the H−O RDF and the first peak in the O−O RDF (Fig. 4b). We emphasise that both fits could most likely be improved by using hand-crafted datasets: the aim of Fig. 4 is not to benchmark specific architectures, but to test what type of practical use can be gained from an MLIP model fitted purely to automatically generated RSS data.

The unique properties of liquid water are usually attributed to its strong hydrogen-bond network[85,86]. Figure 4c shows that both our GAP

and NequIP models, fitted to the same `autoplex`-generated GAP-RSS dataset, can appropriately describe the number of hydrogen bonds per molecule in liquid water in MD simulations, which matches the previously reported range of 3.48 to 3.84 based on different density functionals[87]. The NequIP model predicts a slightly higher average than the original GAP-RSS one, around 3.5, which is closer to the upper end of the range. We note in passing that MD simulations using random-search-based 'ephemeral data-derived potentials' (EDDPs) have been presented recently[88], in that case for hydrogen diffusion in ScH$_{12}$.

Beyond liquid water, we also tested our models by predicting the energies of 54 ice crystal structures (structures taken from ref. 89; Fig. 4d). Here, the results from the GAP-RSS model are highly scattered, whereas the NequIP model shows much improved predictive accuracy. The poor performance of the GAP model could be due to the sparsity of low-density phases in the dataset—which, in turn, would underscore the extrapolation capability of the NequIP model, allowing it to better handle structures that are substantially different from the training data.

The above discussion suggests that the GAP-RSS dataset is not only effective for training a GAP model itself, but is also beneficial for use with other, more complex fitting frameworks. Future work could explore the suitability of `autoplex`-generated (GAP-) RSS datasets as a pre-training task for subsequent fine-tuning of MLIP models, building on our previous study which showed that pre-training NequIP models can enhance not only numerical quality in the low-data regime, but also stability in MD simulations[90].

## Application to chalcogenide memory materials

We finally demonstrate the applicability of `autoplex` to an inorganic material system of 'real-world' interest. For this purpose, we focus on two ternary chalcogenides, viz. Ge$_1$Sb$_2$Te$_4$ and In$_3$Sb$_1$Te$_2$, which are relevant for applications in phase-change memory devices[91–93]. We have recently hand-built a GAP model for compositions along the pseudo-binary line between GeTe and Sb$_2$Te$_3$ (referred to as 'GST' alloys)[26], enabling an accurate description of the amorphous structure (digital 'zero bits') and its crystallisation ('0 → 1'). One of the challenges in modelling amorphous GST is the formation of tetrahedral structural motifs (Fig. 5a), which are relevant for ageing phenomena: the amount of those tetrahedra has been argued to change over time, affecting the stability of the zero bits[94]. However, this hand-crafted GAP model (denoted 'GST-GAP-22') took months to complete, involving multiple runs of domain-specific iterations to cover the structural complexity of the Ge−Sb−Te system[26]. This makes Ge$_1$Sb$_2$Te$_4$ an excellent candidate for testing our automated, RSS-based workflows in computational practice.

Compared to GST, In$_3$Sb$_1$Te$_2$ ('IST' in the following) is a rather unconventional phase-change material, with structural building blocks slightly different from those of GST. The latter alloys structurally resemble their constituent binary phases (viz. GeTe and Sb$_2$Te$_3$), and they can take disordered and defective rocksalt-like metastable structures (see ref. 95 and references therein). The relevant ternary In−Sb−Te compound, viz. In$_3$Sb$_1$Te$_2$, crystallises in a disordered rocksalt-type structure with no substantial amount of cation vacancies[96]. Unlike GST, which does contain such vacancies, IST has all cation sites fully occupied by In atoms (Fig. 5b), with no notable cation disorder. Instead, its structural complexity arises from anion disorder, as Sb and Te share the same substructure. A previous ab initio MD study revealed that amorphous IST exhibits a large number of four-, five-, and six-membered rings[97] (Fig. 5b, right), indicating that its medium-range order is more complex than that of amorphous GST (where four-membered rings are predominant).

In the context of MLIP development, IST is an example of a less widely explored material: there are not as many ab initio MD studies as for GST, nor is there an existing potential model to our knowledge. We argue that instead of manually constructing a training dataset, the user can now use automated approaches, such as the one in `autoplex`, to study less-common functional materials, at least as a starting point.

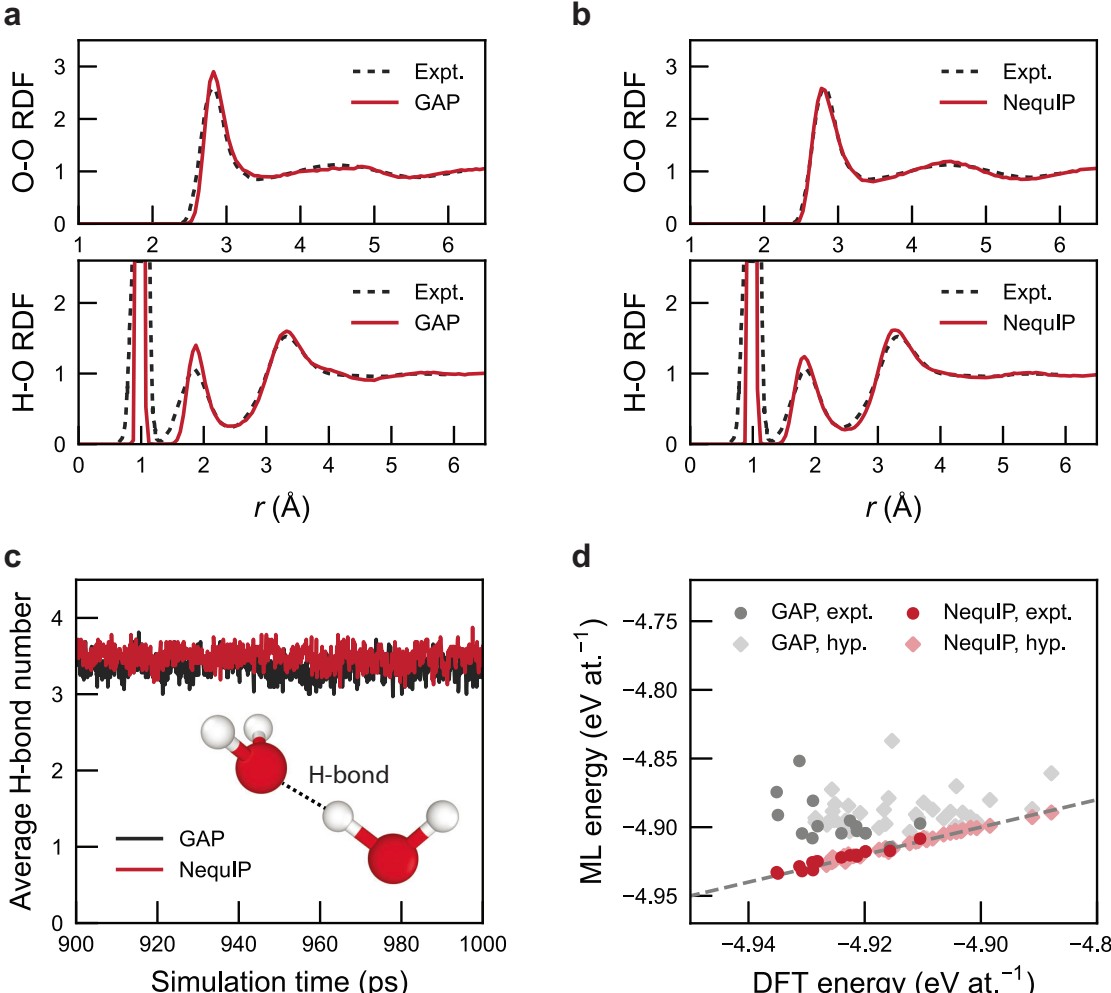

**Fig. 4 | Liquid water and ice polymorphs. a** Radial distribution functions (RDFs) of H–O and O–O pairs comparing experimental data (H–O: ref. [121]; O–O: ref. [122]) with Gaussian approximation potential (GAP) model predictions at 300 K. The symbol $r$ denotes the radial distance from a reference particle. **b** Same for predictions from a NequIP model, fitted to the same training set as the GAP model. **c** Average hydrogen-bond number as a function of simulation time for GAP and NequIP models. The hydrogen-bond criterion is defined as the O–O distance being smaller than 3.5 Å and the O–H–O angle being larger than 140°. **d** Comparison of predicted energies from GAP and NequIP models fitted to the GAP-driven random structure searching (GAP-RSS) dataset, against density functional theory (DFT) energies for various ice polymorphs[89], including those experimentally observed (labelled 'expt.') and those theoretically hypothesised (labelled 'hyp.'). For this test, structures were taken from ref. [89], and re-evaluated with DFT using the revPBE-D3(zero) functional[79]. This plot highlights the improved extrapolation capability of the NequIP model, compared to the initial GAP-RSS one, in capturing energy trends across different phases. Source data are provided as a Source Data file.

Figure 5c–g characterises the performance of GAP-RSS models for both phase-change materials. We first quantified the local structural properties by computing RDFs (Fig. 5c, e), and then calculated ring statistics which allow us to assess the medium-range order in those structures (Fig. 5d, f). The performance of the GAP-RSS models is encouraging, given how fast and inexpensive it has been to fit compared to an existing MLIP for Ge–Sb–Te compounds. Methodologically, the RSS approach in `autoplex` provides structural diversity by sampling from small cells, a strategy supported by previous literature[48,98]. We obtain a robust potential using 46% fewer atomic environments for $Ge_1Sb_2Te_4$ compared to the training dataset in ref. [26], which comprises 49,056 atomic environments for this specific composition (and more for chemically related compounds). More importantly, however, the approach of ref. [26] was based on the expert-guided construction of training datasets, requiring substantial human effort and domain knowledge. In contrast, the RSS methodology enables broad exploration of the potential-energy landscape, requiring much less prior knowledge. In terms of implementation, the `autoplex` framework largely automates the iterative ML-driven RSS process, thereby enhancing the efficiency and ease of applying this

methodology. The results for both phase-change materials show that our approach not only works well for materials with established domain knowledge (GST) but also performs effectively for materials for which so far there is limited domain knowledge (IST).

We finally ran a GAP-driven MD simulation of the crystallisation process in $Ge_1Sb_2Te_4$ (Fig. 5g), corresponding to the SET operation in memory devices ('0 → 1'). We used a kernel similarity metric[99] to quantify the gradual structural ordering process, as in our previous work (ref. [100]). The simulation shows a rapid growth proceeding at the amorphous–crystalline interface, leading to the formation of a largely ordered, defective rocksalt-like structure. We note that this crystallisation simulation using GAP was essentially completed after 350 ps, more quickly than what was seen in ab initio (DFT-based) MD simulations[100] and using the hand-crafted GST-GAP-22 potential[26]. However, the former approach is computationally highly demanding[100], and the latter is a specialised MLIP that has been deliberately trained on 'domain-specific' configurations that correspond to the intermediate steps between fully amorphous and fully recrystallised Ge–Sb–Te materials. Our GAP-RSS based model, used to drive the simulations characterised in Fig. 5g, is trained in a much

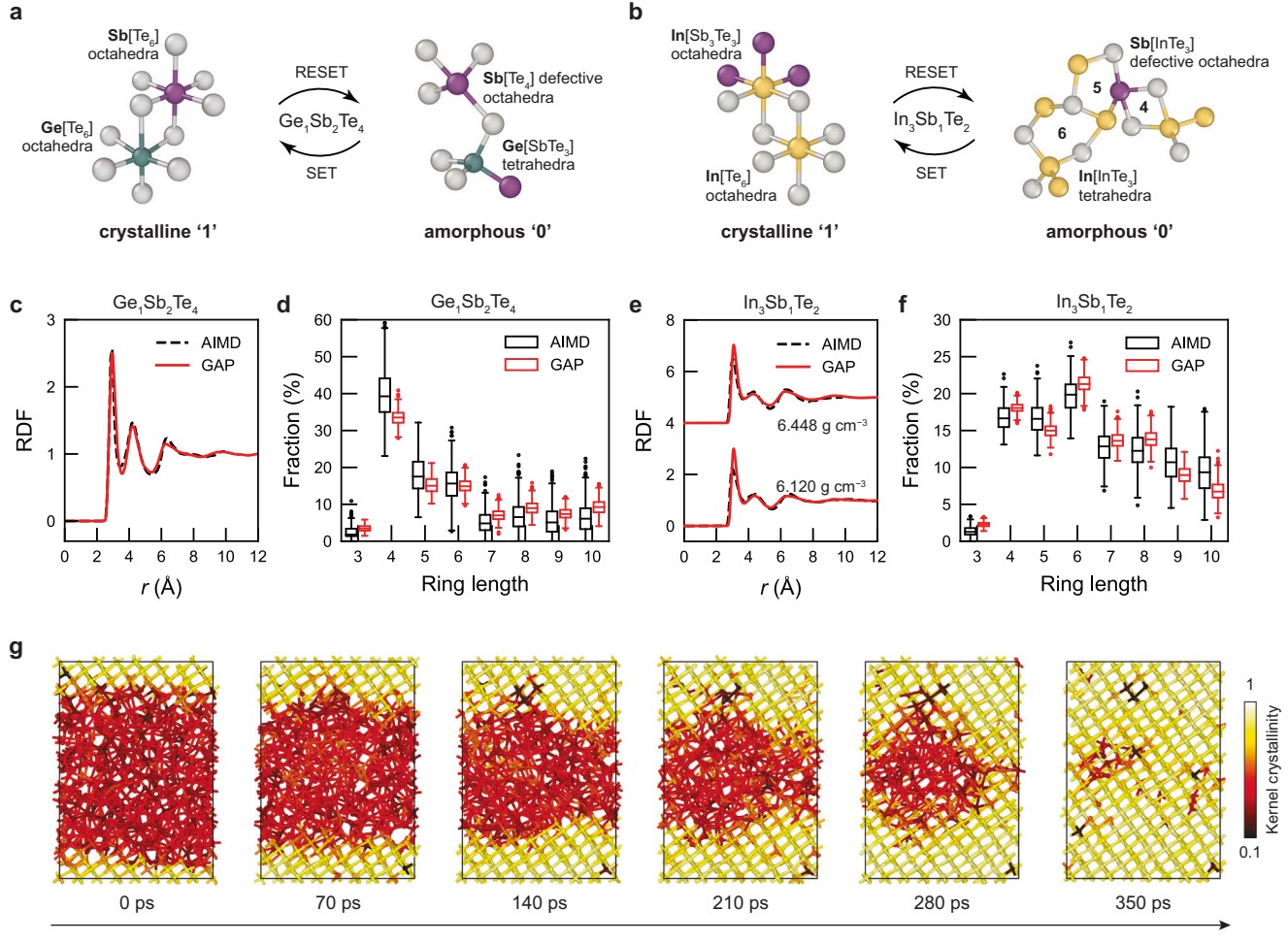

**Fig. 5 | Structural and dynamic properties of phase-change materials.**
**a**, **b** Schematics of the structural transitions for $Ge_1Sb_2Te_4$ and $In_3Sb_1Te_2$, respectively, between crystalline ('1') and amorphous ('0') phases. Atoms are colour-coded as follows: Ge in green, Sb in purple, Te in grey, and In in yellow. (**c**) Radial distribution function (RDF) and (**d**) distribution of shortest-path rings in $a$-$Ge_1Sb_2Te_4$. **e**, **f** Same but for $a$-$In_3Sb_1Te_2$. The symbol $r$ represents the radial distance from a reference particle. All ab initio molecular dynamics (AIMD) data for $a$-$Ge_1Sb_2Te_4$ are sourced from the literature[26]. For $a$-$In_3Sb_1Te_2$, the AIMD-derived RDF data for a density of 6.448 g cm$^{-3}$ are sourced from ref. 97, while the AIMD data at 6.120 g cm$^{-3}$ for the RDF in panel (**e**) and the ring distribution in panel (**f**) are obtained from our own simulations with CP2K[123,124] (Supplementary Note 1). The label 'GAP' in these panels refers to Gaussian approximation potential (GAP) models obtained via random structure searching (RSS). The boxplots in panels (d) and (f) represent the distribution of values across trajectories (see Supplementary Note 1 for statistics). The central line in each box indicates the median, the box spans the inter-quartile range (IQR), and the whiskers extend to 1.5 × IQR. Outliers are shown as individual points. **g** A crystallisation simulation of $Ge_1Sb_2Te_4$ using our GAP-RSS-derived potential, showing the progressive crystallisation over a simulation period of 350 ps. Atoms are colour-coded based on a kernel-based 'crystallinity' measure[99,100]: yellow indicates high crystallinity, and red indicates amorphous regions. Source data are provided as a Source Data file.

simpler way, and using only relatively small-scale configurations—perhaps it could serve as a starting point for constructing subsequent, more complex MLIP training datasets. Further tests for other phase-change memory materials are planned for the future.

## Discussion

Automation is one of the major open challenges in ML potentials for materials and poised to accelerate their development into general mainstream simulation models. We have here shown how iterative exploration and MLIP fitting, initially proposed within the GAP-RSS framework[43], can be automated substantially and at scale by integration with existing software ecosystems. The resulting `autoplex` code is openly available and free to use, and we expect that in the long run, it will develop into a computational ecosystem of its own for the next generation of ML-driven materials modelling.

Although the present study has focused on bulk materials, the ML-driven RSS method is, in principle, well-suited for exploring a broader range of configurations (e.g., clusters, surfaces, and interfaces), as supported by previous applications of the established AIRSS method[101]. Accurately sampling complex dynamical processes, such as reaction pathways and adsorption sites in surface catalysis, remains a common challenge for various structural exploration methods, including RSS and MD, as these processes involve overcoming substantial energy barriers. In this context, the MLIPs produced by our RSS method can provide starting points for further refinement via active learning—an approach that has already proven effective in tackling similar dynamical challenges[40]. Such functionalities could be integrated into `autoplex` in the future.

Our work also contributes to addressing the wider-ranging open question about how MLIPs are best developed and used going forward. Random structure searching provides a core approach to generating robust potentials, and our present results suggest a remarkable amount of stability that can be gained from RSS using small cells alone: therefore, ML-driven iterative RSS appears to emerge as a standard

technique for at least providing a starting point for potential fitting[102]. We have previously pointed out that RSS datasets can constitute useful benchmark tasks for evaluating MLIP models[103,104], and we expect the growing usefulness of this aspect as systematic benchmarking becomes more important in the community.

In the years ahead, we expect that automated approaches to dataset construction—including the one in `autoplex` we have presented here—will play an increasingly important role in the field. With these efforts, and together with the demonstrated capabilities of advanced MLIP fitting architectures[27,28,52,53], universal ML models for atomistic simulations could become widely established—which could make ML-driven modelling the genuine default in the field, just like direct quantum-mechanical modelling has been the default for many years.

## Methods
### Machine-learned interatomic potentials
MLIPs represent a given quantum-mechanical potential-energy surface. The models used herein are based on a local (atom-wise) decomposition of the total energy[15,16],

$$E = \sum_i \varepsilon(\mathbf{q}_i),\tag{1}$$

where the atomic energies, $\varepsilon$, are learned as a function of the atom's local environment, expressed through a general structural descriptor, $\mathbf{q}_i$, and the ML-predicted total energy, $E$, is obtained by summing over the per-atom contributions.

For most of the present study, we used the GAP framework[15], together with the SOAP descriptor[99] to featurise atomic environments. However, our approach is more general, and we have also implemented interfaces to other architectures and codes, including `ACEpotentials.jl`[105], NequIP[19], M3GNet[21], and MACE[20], in the current public version of `autoplex`. We refer to the original literature for details of these methods.

### RSS and iterative fitting
Random structure searching is an established approach to exploring configurational spaces and has seen large success in the context of the AIRSS framework by Pickard and Needs[50,51]. In ref. 43, it was proposed to combine RSS exploration with the fitting of potential-energy surfaces, gradually improving the models through iterative training (which had already been in itself a standard approach using iterative MD[23]) and not requiring any DFT-based relaxation but only single-point DFT evaluations, running all relaxations with GAP models instead. Note that the combination of structural searches and iterative MLIP fitting is not at all restricted to GAP or (AI)RSS—it has been demonstrated for other frameworks as well[44,46].

The approach was extended in ref. 47 by including appropriate selection steps (both structurally and energetically based). Subsequent work introduced the `wfl` software[65]. The latter work is different from ours in that it runs workflows through a custom implementation, whereas `autoplex` interfaces to `atomate2` and its diverse set of DFT-based workflows where possible.

Within `autoplex`, we have incorporated several methodological features. Our framework now supports multiple sets of `buildcell` input parameters, which define the scope of the RSS search and directly impact the diversity of the generated structures. We have also implemented a Hookean repulsion force to prevent atoms from approaching each other too closely, which can help to produce more physically plausible configurations.

Furthermore, if a pre-existing MLIP along with its training dataset is available, our RSS framework can be readily used to generate supplementary data and further refine the potential. One common approach is to combine an existing dataset with configurations generated during the initial RSS stage and use the merged data to train a new MLIP that subsequently drives iterative RSS exploration. Alternatively, a trained MLIP can be used to initiate the first round of RSS iterations, after which the newly generated data may be merged with the original training dataset to improve the model, which is then used in subsequent RSS iterations. Finally, users may opt to generate an RSS dataset from scratch without relying on any prior data or potentials; once the structure searching is complete, this RSS dataset can be merged with pre-existing data for training a new MLIP. More details on the procedures and parameters of the RSS workflows are provided in Supplementary Note 1. The scripts and input parameters for running the workflows are provided in ref. 106.

### Automation
In the present work, we focus on automating RSS and RSS-driven iterative potential fitting. We describe a software implementation that is connected to the `atomate2` ecosystem. `atomate2` is a library of computational materials science workflows that have mostly been developed in the context of the Materials Project[41]. As in `atomate2`, we rely on `jobflow`[107] for writing workflows and `jobflow-remote` or `fireworks`[108] for executing workflows and scheduling computing tasks. For handling DFT inputs and outputs and interfaces with ML potentials, we use `pymatgen`[109] and the Atomic Simulation Environment (ASE)[110].

Specifically, we use existing `atomate2` workflows for automatic DFT computations. For the MLIP fitting part, we provide direct interfaces to relevant software. We further automate the intermediate stage between data generation (typically via DFT) and MLIP fitting.

### Reference data
Reference data were obtained using projector augmented-wave (PAW) potentials[111,112] as implemented in the Vienna Ab Initio Simulation Package (VASP)[112,113], version 6.3.2. To describe the effects of exchange and correlation, we used the Perdew–Burke–Ernzerhof (PBE) functional[74], its revised parameterisation for solids (PBEsol)[114], revPBE[115] with the D3 dispersion correction[79], as well as the Strongly Constrained and Appropriately Normed (SCAN) functional[75], depending on the specific material system to be studied. The DFT settings were similar to previous work for silicon[47], Ti–O[116], $SiO_2$ (ref. 25), water[29], and phase-change materials[26] (here using the same settings for GST and IST), and adapted where appropriate. Structures were visualised using OVITO[117].

### Reporting summary
Further information on research design is available in the Nature Portfolio Reporting Summary linked to this article.

## Data availability
Data supporting this work are openly available via GitHub at https://github.com/autoatml/papers-autoplex-rss, with a permanent version available via Zenodo at https://doi.org/10.5281/zenodo.15720026 (ref. 106) and https://doi.org/10.5281/zenodo.15258384 (ref. 118). Source data are provided with this paper.

## Code availability
The `autoplex` software is openly available at https://github.com/autoatml/autoplex. The code is under ongoing development; a copy of the version used for the results presented herein (`v0.0.7`) is deposited in the Zenodo repository (https://doi.org/10.5281/zenodo.14169361, ref. 119).

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

## Acknowledgements

We thank L.-B. Paşca and L. Wu for help with early tests. This work was supported through a UK Research and Innovation Frontier Research grant [grant number EP/X016188/1]. J.D.M. acknowledges funding from the EPSRC Centre for Doctoral Training in Inorganic Chemistry for Future Manufacturing (OxICFM), EP/S023828/1. We are grateful for computational support from the UK national high performance computing service, ARCHER2, for which access was obtained via the UKCP consortium and funded by EPSRC grant ref EP/X035891/1, as well as through a separate EPSRC Access to High-Performance Computing award[120]. Additionally, we acknowledge the Gauss Centre for Supercomputing e.V. (www.gauss-centre.eu) for funding this project by providing generous computing time on the GCS Supercomputer SuperMUC-NG at Leibniz Supercomputing Centre (www.lrz.de) (project pn73da) that enabled testing of the implementations in autoplex.

## Author contributions

Y.L. and J.D.M. developed the RSS automation code in autoplex. Y.L., C.E., N.L.F., A.A.N., and J.G. are the core autoplex code developers and maintainers at the time of this writing. Y.L., J.L.A.G., N.L.F., and Y.Z. carried out numerical experiments. J.G. and V.L.D. designed and supervised the research. Y.L. and V.L.D. drafted the manuscript, and all authors contributed to the final version.

## Competing interests

The authors declare no competing interests.
