## [Transparent Peer Review file · Nature Communications]

An automated framework for exploring and learning potential-energy surfaces

Corresponding Author: Professor Volker Deringer

Version 0:

Reviewer comments:

Reviewer #1

(Remarks to the Author)

In this manuscript, the authors present an automated framework for constructing machine-learned interatomic potentials (MLIPs) with high predictive accuracy and demonstrate its utility through several examples. I concur with the authors' assessment that, while architectures for high-accuracy MLIPs are now well-established, the primary bottleneck lies in obtaining high-quality training data. Although methodologies for collecting such data are advancing, the current practice still heavily relies on manual effort and domain-specific expertise, as the authors rightly point out.

Given the broad applicability of MLIPs across various research fields, many researchers and engineers are expected to find the proposed framework and code highly valuable for developing and applying MLIPs to their own projects. Thus, I believe the manuscript will attract considerable interest from the community. The paper is well-structured, with its arguments clearly presented and supported by diverse examples, making it a compelling contribution. For these reasons, I strongly recommend its publication without delay.

That said, there is room for minor improvements to enhance its accessibility, particularly for non-expert readers. For example, the authors refer to their previous work (Ref. 26), where MLIPs were created, and claim that the present study achieves more efficient automated training data collection. However, it remains unclear to what extent the number of DFT calculations required to obtain a sufficient level of predictive accuracy has been reduced compared to the previous work. Alternatively, if the primary benefit lies solely in the automation of the data collection process without a significant reduction in the number of DFT calculations, this should be clarified. Including specific comparisons or descriptions in this context would improve the manuscript's clarity and impact. I do not, however, request any additional computations to be performed for this purpose.

Additionally, researchers who already possess datasets for constructing MLIPs may be particularly interested in leveraging the proposed framework to efficiently gather supplemental data or refine their MLIPs. While this approach appears feasible in principle, a brief explanation of whether the current version of the code allows for this functionality would make the paper even more appealing.

In summary, while the manuscript is already well-prepared for publication, addressing these minor suggestions could further broaden its accessibility and impact.

(Remarks on code availability)

I have accessed the URL and reviewed the 'README' file. However, I have not examined the code in detail or attempted to run it.

Reviewer #3

(Remarks to the Author)

The authors present the Autoplex framework and evaluate the results obtained using this framework to iteratively improve machine learning potential models for atomistic simulation. This framework provides a much-needed tool for the community to construct models for their atomistic simulation needs. Additionally, the authors demonstrate the significant advantages of

using their software to develop models for specific use cases. I recommend this article for publication, as it can significantly impact researchers' ability to build and use their own machine learning interatomic potential models.

(Remarks on code availability)

Reviewer #4

(Remarks to the Author)

The manuscript presents Autoplex, an automated framework for developing machine-learned interatomic potentials (MLIPs) using an iterative random structure search and active learning approach. The framework integrates AIRSS with high-throughput workflows to explore configurational spaces and refine MLIPs. The study demonstrates applications across bulk materials, including Si, Ti-O systems, SiO₂ polymorphs, water, and phase-change materials. The authors claim that Autoplex provides a scalable and efficient approach for automating PES exploration and MLIP development.

Overall, I believe that the Autoplex framework is a significant advancement in the automation of ML potential fitting. However, while the approach is effective for bulk systems, the reliance on AIRSS raises concerns regarding its applicability to low-dimensional systems such as surface reactions, interfaces, and defects. AIRSS is primarily designed for bulk crystal exploration and does not efficiently sample adsorption sites, reaction pathways, or local structural distortions commonly found in surface and interfacial environments. The lack of constraints in AIRSS-generated structures may lead to unphysical configurations when applied to slab models, where vacuum gaps and fixed layers must be considered. Revising the manuscript to acknowledge these limitations and discuss potential solutions will provide a more balanced perspective on the framework's capabilities and future improvements.

(Remarks on code availability)

Version 1:

Reviewer comments:

Reviewer #1

(Remarks to the Author)

(Remarks on code availability)

Reviewer #4

(Remarks to the Author)

The revised manuscript now candidly states that the present benchmarks are limited to bulk systems and that surfaces, interfaces, and reaction pathways remain open challenges, while the rebuttal outlines how autoplex-generated MLIPs can be refined through active-learning loops or coupled to forthcoming surface-specific workflows. With these additions the authors have acknowledged the methodological boundary I raised and have introduced a realistic path for users who wish to tackle low-dimensional problems, thereby removing the risk of readers over-generalising the current results. Thus I support publication, suggesting a brief mention of the limitation in the Introduction as an optional revision.

(Remarks on code availability)

Response to Reviewers' Comments for "An automated framework for exploring and learning potential-energy surfaces"

We thank all reviewers for their thoughtful and positive evaluation of our work. We are grateful for their recognition of the significance of the `autoplex` framework, and we value their constructive suggestions which have helped us to improve the manuscript further.

The main changes are briefly summarised below. The first point addresses specific reviewer comments, while the second and third points highlight additional improvements we made to further strengthen the manuscript.

- We further discuss the advantages of our current approach compared to previous hand-crafted training datasets and highlight current limitations and potential strategies for future improvement in the revised Discussion section.
- Since the initial submission, we have improved the accuracy of our GAP-RSS models for describing liquid water (Fig. 4) by changing the DFT approach used to label the training data to revPBE-D3(zero).
- We now provide additional computational details in the Methods section and in a brief Supplementary Information document.

In the following, we reproduce the reviewers' "Remarks to the Author", with our point-by-point responses shown in **blue** and action taken indicated in **red**.

Reviewer #1

In this manuscript, the authors present an automated framework for constructing machine-learned interatomic potentials (MLIPs) with high predictive accuracy and demonstrate its utility through several examples. I concur with the authors' assessment that, while architectures for high-accuracy MLIPs are now well-established, the primary bottleneck lies in obtaining high-quality training data. Although methodologies for collecting such data are advancing, the current practice still heavily relies on manual effort and domain-specific expertise, as the authors rightly point out.

Given the broad applicability of MLIPs across various research fields, many researchers and engineers are expected to find the proposed framework and code highly valuable for developing and applying MLIPs to their own projects. Thus, I believe the manuscript will attract considerable interest from the community. The paper is well-structured, with its arguments clearly presented and supported by diverse examples, making it a compelling contribution. For these reasons, I strongly recommend its publication without delay.

Response: We thank the reviewer for the positive feedback on our manuscript.

That said, there is room for minor improvements to enhance its accessibility, particularly for non-expert readers. For example, the authors refer to their previous work (Ref. 26), where MLIPs were created, and claim that the present study achieves more efficient automated training data collection. However, it remains unclear to what extent the number of DFT calculations required to obtain a sufficient level of predictive accuracy has been reduced compared to the previous work. Alternatively, if the primary benefit lies solely in the automation of the data collection process without a significant reduction in the number of DFT calculations, this should be clarified. Including specific comparisons or descriptions in this context would improve the manuscript's clarity and impact. I do not, however, request any additional computations to be performed for this purpose.

Response: We thank the reviewer for this helpful suggestion. In previous work, a manually constructed training set was developed to cover the entire GeTe–Sb₂Te₃ quasi-binary compositional line, whereas our focus here is solely on Ge₁Sb₂Te₄. Within this context, the earlier dataset included a subset of 286 Ge₁Sb₂Te₄ structures, each averaging 172 atoms, resulting in a total of 49,056 atomic environments. (We note that additional structures with similar compositions, such as Ge₁Sb₄Te₇ and Ge₂Sb₂Te₅, also exhibit comparable local atomic environments and may help to improve the accuracy of the description for Ge₁Sb₂Te₄.) With a smaller number of atomic environments in the present work, the RSS-based method already leads to a robust potential, by providing structural diversity through sampling from many small cells.

The most important difference, however, is not just in the number of DFT computations required, but in the human effort: the earlier approach in Ref. 26 relied heavily on iterative, expert-guided construction of training datasets, requiring significant amounts of time and domain expertise—and, as we mention on p. 16, yielding a higher-quality potential overall (as measured by the crystallisation speed). In contrast, the RSS methodology enables a broad exploration of the potential-energy landscape without relying on prior knowledge of specific atomic arrangements, aside from defining sensible input parameters—and the automation in `autoplex` improves the efficiency and ease of use.

Action taken: We added the following text to further compare the RSS approach as implemented in `autoplex` to the previous, hand-crafted potential of Ref. 26:

“Methodologically, the RSS approach in `autoplex` provides structural diversity by sampling from small cells, a strategy supported by previous literature^{49,103}. We obtain a robust potential using 46% fewer atomic environments for $\text{Ge}_1\text{Sb}_2\text{Te}_4$ compared to the training dataset in Ref. 26, which comprises 49,056 atomic environments for this specific composition (and more for chemically related compounds). More importantly, however, the approach of Ref. 26 was based on the expert-guided construction of training datasets, requiring substantial human effort and domain knowledge. In contrast, the RSS methodology enables broad exploration of the potential-energy landscape, requiring much less prior knowledge. In terms of implementation, the `autoplex` framework largely automates the iterative ML-driven RSS process, thereby enhancing the efficiency and ease of applying this methodology.” (pp. 14–16)

Additionally, researchers who already possess datasets for constructing MLIPs may be particularly interested in leveraging the proposed framework to efficiently gather supplemental data or refine their MLIPs. While this approach appears feasible in principle, a brief explanation of whether the current version of the code allows for this functionality would make the paper even more appealing.

In summary, while the manuscript is already well-prepared for publication, addressing these minor suggestions could further broaden its accessibility and impact.

Response: Indeed, our code already supports the functionality mentioned by the reviewer, which we agree is certainly worth emphasising in the revised manuscript. Specifically, users can readily employ an existing MLIP model to drive RSS exploration, and then combine the newly generated RSS datasets with previous training data to refine their potentials. Alternatively, users can generate RSS datasets entirely from scratch and subsequently merge them with existing datasets for training a new potential.

Action taken: To clarify this point, we extended the online documentation of `autoplex`: see https://autoatml.github.io/autoplex/user/rss/flow/quick_start/start.html. We also extended the Methods section of the manuscript accordingly (p. 19).

Reviewer #3

The authors present the Autoplex framework and evaluate the results obtained using this framework to iteratively improve machine learning potential models for atomistic simulation. This framework provides a much-needed tool for the community to construct models for their atomistic simulation needs. Additionally, the authors demonstrate the significant advantages of using their software to develop models for specific use cases. I recommend this article for publication, as it can significantly impact researchers' ability to build and use their own machine learning interatomic potential models.

Response: We thank the reviewer for the positive evaluation.

Reviewer #4

The manuscript presents Autoplex, an automated framework for developing machine-learned interatomic potentials (MLIPs) using an iterative random structure search and active learning approach. The framework integrates AIRSS with high-throughput workflows to explore configurational spaces and refine MLIPs. The study demonstrates applications across bulk materials, including Si, Ti-O systems, SiO₂ polymorphs, water, and phase-change materials. The authors claim that Autoplex provides a scalable and efficient approach for automating PES exploration and MLIP development.

Overall, I believe that the Autoplex framework is a significant advancement in the automation of ML potential fitting. However, while the approach is effective for bulk systems, the reliance on AIRSS raises concerns regarding its applicability to low-dimensional systems such as surface reactions, interfaces, and defects. AIRSS is primarily designed for bulk crystal exploration and does not efficiently sample adsorption sites, reaction pathways, or local structural distortions commonly found in surface and interfacial environments. The lack of constraints in AIRSS-generated structures may lead to unphysical configurations when applied to slab models, where vacuum gaps and fixed layers must be considered. Revising the manuscript to acknowledge these limitations and discuss potential solutions will provide a more balanced perspective on the framework's capabilities and future improvements.

Response: We appreciate the positive evaluation as well as the critical comment. Indeed, AIRSS has been developed for predicting not only bulk crystal structures but also broader regions of configurational space, including clusters, defects in solids, surfaces, and interfaces (see <https://airss-docs.github.io/> and relevant literature^{R1,R2}). Accurate sampling of reaction pathways and surface catalytic phenomena remains a challenging task not only for RSS, but also for other methods such as (AI)MD simulations.

The advantage of our ML-driven RSS approach lies in its ability to comprehensively sample the potential-energy surface, facilitating the development of MLIPs that can describe crystals (across various phases and stoichiometric ratios), liquids, and amorphous solids, as shown in the present work. These potentials can then serve as robust starting points for further refinement: *e.g.*, through active-learning strategies aimed at capturing rare dynamic behaviours in chemical reactions.

We emphasise that `autoplex` is designed as a comprehensive, integrated platform. In the present work, we specifically focus on the RSS-based workflow; however, other current or future developments include additional functionality such as, *e.g.*, phonon workflows focused on crystalline structures. To date, the phonon workflow is already available in the public version, while other developments are expected to be released in the future.

References:

- R1. Schusteritsch, G. & Pickard, C. J. Predicting interface structures: From SrTiO₃ to graphene. *Phys. Rev. B* **90**, 035424 (2014).
- R2. Karasulu, B. et al. Accelerating the prediction of large carbon clusters via structure search: Evaluation of machine-learning and classical potentials. *Carbon* **191**, 255–266 (2022).

Action taken: We extended the discussion of the limitations of the current framework and suggest possible directions for improvement:

“Although the present study has focused on bulk materials, the ML-driven RSS method is, in principle, well-suited for exploring a broader range of configurations (e.g., clusters, surfaces, and interfaces), as supported by previous applications of the established AIRSS method¹⁰⁴. Accurately sampling complex dynamical processes, such as reaction pathways and adsorption sites in surface catalysis, remains a common challenge for various structural exploration methods, including RSS and MD, as these processes involve overcoming substantial energy barriers. In this context, the MLIPs produced by our RSS method can provide starting points for further refinement via active learning—an approach that has already proven effective in tackling similar dynamical challenges⁴⁰. Such functionalities could be integrated into `autoplex` in the future.” (p. 17)

Reviewer #4

The revised manuscript now candidly states that the present benchmarks are limited to bulk systems and that surfaces, interfaces, and reaction pathways remain open challenges, while the rebuttal outlines how autoplex-generated MLIPs can be refined through active-learning loops or coupled to forthcoming surface-specific workflows. With these additions the authors have acknowledged the methodological boundary I raised and have introduced a realistic path for users who wish to tackle low-dimensional problems, thereby removing the risk of readers over-generalising the current results. Thus I support publication, suggesting a brief mention of the limitation in the Introduction as an optional revision.

Response: We appreciate the positive evaluation and the good suggestion. We have now included a brief discussion of the limitations of the current work in the Introduction.

Action taken: We added the following statement on p. 3:

“Note that our benchmarks here are restricted to bulk systems, whereas surfaces, interfaces, and reaction pathways remain important and challenging frontiers for future extension.”